# The Effects of Different Diets and Transgenerational Stress on *Acyrthosiphon pisum* Development

**DOI:** 10.3390/insects10090260

**Published:** 2019-08-21

**Authors:** Daniel Pers, Allison K. Hansen

**Affiliations:** Department of Entomology, University of California, Riverside, CA 92507, USA

**Keywords:** *Acyrthosiphon pisum*, pea aphid, nymphal instars, nymphal development, plant-rearing, artificial liquid diet, transgenerational stressors, developmental delay, siphunculus

## Abstract

Despite the fact that sap-feeding hemipterans are major agricultural pests, little is known about the pea aphid’s (*Acyrthosiphon pisum*) nymphal development, compared to other insect models. Given our limited understanding of *A. pisum* nymphal development and variability in the naming/timing of its developmental events between different environmental conditions and studies, here, we address developmental knowledge gaps by elucidating how diet impacts *A. pisum* nymphal development for the LSR1 strain when it develops on its universal host plant (*Vicia faba*), isolated leaves, and artificial diet. Moreover, we test how plant age and transgenerational stressors, such as overcrowding and low plant vigor, can affect nymphal development. We also validate a morphological method to quickly confirm the life stage of each nymphal instar within a mixed population. Overall, we found extremely high variation in the timing of developmental events and a significant delay in nymphal (~5–25-h/instar) and pre-reproductive adult (~40-h) development when reared on isolated leaves and artificial diets, compared to intact host plants. Also, delays in development were observed when reared on older host plants (~9–17-h/event, post 2nd instar) or when previous generations were exposed to overcrowding on host plants (~20-h delay in nymph laying) compared to controls.

## 1. Introduction

Sap-feeding insects within the insect order Hemiptera are major agricultural pests and vector a diversity of plant diseases [1]. While plant-feeding Hemiptera consists of many diverse lineages of taxa, aphids often serve as a model system for genomics. *Acyrthosiphon pisum* specifically, was the first Hemipteran genome to be sequenced and it is used as a significant reference genome for Hemiptera [2,3]. However, while *A. pisum* may be an important sap-feeding pest model, very little is known about its nymphal development compared to insect models that belong to other insect orders. Previous developmental studies in *A. pisum* have primarily focused on polyphenisms observed in the adult life stages; male vs. female, alate vs. apterous morphs, and sexual vs. asexual reproduction [4,5,6,7,8,9,10,11,12,13,14,15,16]. While understanding the development of these polyphenisms is extremely important in aphids, having a deeper understanding of nymphal development is also important because while these phenotypes are visible in adult life stages, patterning and fate specification occur during embryonic and nymphal life stages.

Due to our limited understanding of *A. pisum* nymphal development, a major limitation of experiments that use nymphal life stages is that they tend to primarily acknowledge the aphid’s age after viviposition instead of the actual developmental life stage of the aphid (i.e., 2 day old instead of 2nd instar) [17,18,19,20,21,22,23], or combine all nymphal life stages into one pre-reproductive period [24,25,26,27,28,29]. As a consequence, if the aphid’s developmental stage is unknown, it becomes difficult to compare results of nymphal life stages between different studies, experimental conditions, and other aphid species. In addition, the few studies that provide life cycle tables describing the age/time of each nymphal instar are generally determined for non-LSR1 aphid strains that are reared on host plants under specific abiotic conditions [24,25,26,27,28,29,30,31,32,33,34,35,36,37,38,39,40,41]. Similar data has not been generated for the LSR1 strain, the reference aphid genome, on the commonly used universal host plant, *V. faba*, under standard laboratory rearing conditions [2].

Many experiments involving immature life stages of *A. pisum* use either artificial diet and/or isolated leaf sections and as a result, it is unknown how these manipulated feeding systems can influence aphid development compared to more natural whole plant experimental designs [25,27,31,35,39,42,43,44,45,46,47,48,49]. The advantage of using artificial diets in nutritional studies is that you can manipulate specific metabolites of interest and use the diet as a vehicle to administer drugs and other chemical agents to the aphid [18,25,27,35,39,42,43,44,46,47,48,49]. The disadvantage however is that when *A. pisum* is reared on an artificial diet, they generally take longer to develop, weigh less, and lay far fewer offspring when compared to aphids reared on host plants [35,39,45,48,50]. This sub-optimal performance on artificial diets was also observed in other aphid species, suggesting that these diets are lacking unknown components, and/or are affecting feeding behavior when compared to aphids feeding on intact host plants [51]. In turn, elucidating how nymphal development varies across different experimental diet treatments would be extremely useful for the research community.

In addition to different diet treatments, other environmental factors may modify *A. pisum* nymphal development rates. For example, the winged (alate) morph of female aphids takes longer to develop, and they lay smaller, and fewer, offspring compared to apterous morph (non-winged) clones [4,12,13]. The emergence of alate morph offspring is known to be caused by environmental cues (stressors) detected in the previous generation of aphids. These environmental cues include an increase in aphid-aphid interactions (overcrowding), aphid-predator interactions, and/or a suboptimal host plant [4,5,12,13]. These stressors are perceived and processed by the mother and trigger changes to the daughters, allowing the next generation to develop wings and the ability to fly to a more optimal environment [4,5,12,13]. By understanding the breadth and limits of these transgenerational cues/changes, a greater degree of control can be achieved when setting up and interpreting future studies.

In this study, we address developmental knowledge gaps in elucidating how diet and transgenerational stressors impact aphid development by measuring *A. pisum* nymphal development for the LSR1 strain when it develops on its universal host plant (*Vicia faba*), in addition to other common experimental conditions frequently used by the research community, such as isolated leaf samples and artificial diet. Second, we highlight morphological methods that can be quickly, and easily used by the research community to confirm the developmental life stage of individual aphids even within mixed aged populations. Third, we document how plant age and crowding in previous maternal generations can impact nymphal development.

## 2. Materials and Methods

### 2.1. Aphid Lines

Three independent sub-lines (Fava1, Fava2, Fava3) of *Acyrthosiphon pisum* (LSR1 strain) were reared on fava bean plants (*Vicia faba*) of similar developmental stages (~4–6 whorl stage, ~2–3 weeks post germination) at the conditions described in Kim et al. [52]. All rearing occurred in climate-controlled Intellus Ultra controller Percival incubators (Percival Scientific, Inc, Perry, IA, USA) at 20 °C with a 16:8 h day-night cycle. These sub-lines were isolated, reared, and maintained in this way for over 150 generations.

### 2.2. Developmental Trials on Three Different Diets

For all trials, unless otherwise stated, host plant age and the relative number of individuals on host plants for each sub-line were carefully monitored and controlled for more than two generations prior to the start of each diet treatment trial. Specifically, prior to the start of trials, 5–10 reproductive, adult parthenogenic female clones from each sub-line were transferred to a 4–5 whorl stage *V. faba* plant and allowed to lay nymphs for 48 h. After 48 h, the adults were removed from the plant and the young nymphs were allowed to develop on the plant until adulthood. Once these nymphs became reproductive (~9–10 days), 10–20 reproductive adults from each sub-line were transferred to a new 4–5 whorl stage *V. faba* plant and allowed to lay nymphs for 48 h. Again, after 48 h, the adults were removed from the plant and the young nymphs were allowed to develop on the plant until adulthood. Once these nymphs became reproductive (~9–10 days), ~70 reproductive adult clones from each sub-line were transferred to a new 4–5 whorl stage *V. faba* plant and allowed to lay nymphs for 8 h. After 8 h, the adults were removed.

Newly laid nymphs (0–8 h; 4.0 ± 4.0 h), whose mothers and grandmothers were reared in controlled conditions (see above), were randomly sub-divided into three diet treatments (Plant, Leaf, and Diet). For the leaf treatment, approximately 20 nymphs were removed and transferred to a rearing arena containing a leaf embedded in agar (Leaf, Figure 1A–B). For the diet treatment, ~20 nymphs were removed and transferred to a rearing arena containing an artificial diet (Diet, Figure 1C–E). For the plant treatment, 20+ nymphs were left on the *V. faba* plant that they were viviposited on (Plant). Diet treatments were conducted for all three aphid sub-lines (Fava1, Fava2, Fava3), creating three biological replicates per treatment. Nymphs developed on the diet treatments for ~2 weeks.

Approximately every 1–4 h, 24h per day, all nymphs were observed and noted if a molting event had occurred. For each diet treatment and biological replicate, the number of nymphs observed molting and the time (hours post viviposit) were recorded and the molts were physically removed from the rearing environment with a thin paintbrush. These observations were repeated until all nymphs completed four molting events. After the 4th molt, aphid observation continued; however, instead of scoring for a molt, the developmental event of interest was reproductive maturity, observed through the proxy of nymph laying. Again, at each time point, the number of mothers observed giving birth and the time (hours post their own viviposit) were recorded and both the mother and nymph were physically removed from the rearing environment. Additionally, it was recorded if the mother was a winged (alate) or non-winged (apterous) morph. Moreover, the death and/or escape of an individual nymph was recorded as well throughout the trials.

Whole Fava Plant Diet Treatment (Plant): Fava bean (*Vicia faba*) plants were grown by planting 1–2 “Bean Seeds (Organic Fava)-Broad Windsor” (Eden Brothers, Arden, NC, USA) in a pot (4 in. diameter) full of Master’s Pride Professional Potting Soil (EB Stone & Son, Inc., Suisun City, CA, USA). Plants were grown in a climate-controlled growth room (20–25 °C, 16:8 h light-dark cycle) for ~2–3 weeks, watering as needed.

Isolated Fava Leaf Diet Treatment (Leaf): A 1% solution of bacteriological agar (VWR, Radnor, PA) in de-ionized water, was poured into sterilized petri dishes (Fisher Scientific, Waltham, MA, USA). A total of 1–2 leaves from the oldest whorl of a 4–5 whorl stage *V. faba* plant were gently embedded into the partially cooled agar and allowed to solidify at room temperature. Plates were stored, inverted, at 4 °C overnight prior to use.

A rearing arena was created by cutting a 4 × 4 × 4 cm triangle into a Petri dish lid with a soldering iron (Figure 1A–B). A 110 mm diameter piece of filter paper was glued to the inside of the Petri dish lid (to prevent excess humidity and mold growth). Aphids were transferred to the surface of the agar embedded leaf and covered with the filter paper-coated Petri dish lid. The perimeter of the Petri dish was then sealed with parafilm to prevent aphids from escaping and desiccating. A fresh leaf-embedded agar plate was provided every 24 h.

Artificial Diet Treatment (Diet): Amino acids, vitamins, salts/buffers/sterols, and trace metals were dissolved in a 0.5 mol sucrose solution, in de-ionized water, at the concentrations listed in the Appendix A (Appendix A, [19,53]). Once all compounds were dissolved, 1.25 mL of 1.75 mol KOH was added. Additional KOH was added slowly over a 1–3 h period, until the solution reached a pH of 7.5. De-ionized water was added to bring the volume of the solution up to 100 mL. The solution was vacuum-filter sterilized, aliquoted, and stored at −80 °C.

Two 24-well plates (Corning, Corning, NY, USA) were sandwiched together to create a rearing arena (Figure 1C–E). For one of the plates, a pin sized hole was made in the bottom of each well. A rectangle of fine mesh was then glued to the base of this 24-well plate. For the second plate, each well was filled with 500 uL of artificial diet (spiked with food coloring 1:1000 to ensure that all aphids included in the study were feeding). A rectangle of parafilm (approximately the same size as the 24-well plate) was stretched over the diet filled plate. A tight seal was important to prevent the diet from leaking out of the wells and to ensure that the aphids could pierce their stylets through the film. Aphids (1–2) were transferred into each well of the first plate, and then covered by inverting the second plate over the first, so that the aphids were trapped between the bottom of the well/mesh on plate 1 and the parafilm layer of plate 2. Once aligned, 3–4 rubber bands held the two plates together. The artificial diet was replaced with a fresh aliquot every 24 h, and the 24-well plates containing the diet were replaced every 48 h to prevent microbial growth.

High mortality (40+% as 1st instar, 100% by 3rd instar) was initially observed in Diet-reared aphids. A technical replicate (~20, 1st instar nymphs from the same mothers) was set up the next day. High mortality (88+% as 1st instar, 100% by 3rd instar) was observed in this second technical replicate. Protocol optimization and literature review revealed that aphid nymphs have a much higher survival rate when transferred to artificial diet as 2nd instar nymphs compared to 1st instar nymphs [18,23]. To increase survivorship of aphids when feeding on artificial diet, two additional technical replicate trials were set up sequentially, 8 h apart from each other, following the protocol described above, except instead of transferring ~20 nymphs after 8 h, nymphs were allowed to develop to second instar (after their first molt) on the plant before they were transferred to the Diet treatment (~60 h).

### 2.3. Effects of Plant Age

Reproductive female clones (~30 individuals) from one of the sub-lines (Fava3) were transferred to a 7–8 whorl stage *V. faba* plant and were allowed to lay nymphs for 3 h. After 3 h, 25 nymphs were laid, and the adult females were removed. Newly laid nymphs (0–3 h; 1.5 ± 1.5 h) were allowed to develop on the plant for ~2 weeks. Nymphs were observed and noted if a molting/laying event had occurred every 1–4 h (as described above). Other than the age of the host plant (~1 week older), all other conditions were kept identical to those described above.

### 2.4. Effects of Transgenerational Stress on Aphid Development

Unlike previous experiments, reproductive female clones from each sub-line were exposed to the following unfavorable conditions (see below) for two generations in an attempt to induce stressors such as crowding and low plant vigor on aphid clones. Here, we define stress as the aphid’s developmental response to a suboptimal rearing environment, caused by overcrowding. Specifically, the population was allowed to grow uncontrolled, creating an overcrowded environment. Under controlled conditions, ~10–20 reproductive females are reared on two plants (low initial aphid density [5]). Those aphids and their offspring then feed on the plant for approximately 2 weeks, resulting in a final mixed-age population of ~100 aphids. After two weeks, ~10–20 new reproductive females are transferred to two un-infested plants to begin a new population. In the uncontrolled, overcrowded environment, a new sub-population is not created after two weeks. Instead of removing 10–20 reproductive females to start a new colony, all progeny (~100 aphids) are left on the original plants and allowed to reproduce, resulting in ~1000 aphids total in the following generation. After less than two weeks of aphid feeding, the *V. faba* plants in these experiments changed in coloration from green to yellowish, and there was a reduction in stem and branch turgor pressure. Plant vigor was maintained in the control treatments by keeping aphid populations at a low density and by providing un-infested plants of a specific age (4–5 whorl stage) when reproductive females were transferred. The stress induced by increased overcrowding and low plant vigor can be observed by an increase in the number of winged morphs and yellowish-white colored (unhealthy) aphids [13]. A majority of aphids that developed on the “stressed” plant treatment for two generations were yellowish-white and winged.

After 2 generations under stressful conditions, reproductive female clones from each sub-line were transferred to a 4–5 whorl stage *V. faba* plant and allowed to lay nymphs under low aphid density conditions as described above and under Developmental Trials on Three Different Diets. Once these nymphs became reproductive adults (~9–10 days), ~12 reproductive female clones from each sub-line were transferred to a new 4–5 whorl stage *V. faba* plant and allowed to lay nymphs for 6 h. Newly laid nymphs (0–6 h; 3.0 ± 3.0 h) were allowed to develop on the plant for ~2 weeks. Nymphs were observed and noted if a molting/laying event had occurred every 1–4 h (as described above).

### 2.5. Statistical Analysis of Developmental Events

Molting time was determined to be the number of hours from viviposit until the observance of a molting event. Nymph laying was determined to be the number of hours from its own viviposit to the first nymph that it viviposited. Note that times are continuous from viviposit to each developmental event and are not reset at each event. Molting events were used to signify the transition from one instar nymphal stage to the next instar nymphal stage. Nymph laying was used to signify the transition from a young pre-reproductive adult stage to an older reproductive adult stage.

Event times for all stages from each sub-line replicate and rearing condition were reported as box and whisker plots using the Quartile Calculation: exclusive median formula within Excel, and all statistical tests were conducted using its Data Analysis plugin (Microsoft Excel, Version 16.26). The average, minimal, and maximum times, standard deviation, and range of each event were also calculated.

Single Factor ANOVA tests calculated variation in developmental timing within sub-line biological replicates for each diet treatment/rearing condition at each developmental stage. Two Factor w/ Replication ANOVA tests calculated variation between diet treatments/rearing conditions at each developmental stage (Replication = sub-line replicate). In order to utilize a Two Factor ANOVA, outlier data points were removed from the overall data sets, then randomly selected subpopulations were generated for each data set in order to assign equal sample sizes per stage between conditions. A *p*-value less than or equal to 0.05 signifies significantly different populations due to variation between sample comparisons (diet treatments/rearing conditions), within column replicates (sub-line biological replicates), and/or because of a combined effect of the two. If a group of populations is determined to be significantly different, the Bonferroni approach and post hoc t-test was used to determine which individual population is different from the rest through pairwise comparisons (2 tail, unequal variance, *p* < 0.05).

Two Factor w/ Replication ANOVA tests also calculated variation in developmental timing between morphs (non-winged, winged) at the final developmental stage (Replication = rearing conditions). Again, to utilize a Two Factor ANOVA, outlier data points were removed from the overall data sets, sub-line replicates were combined, and then randomly selected subpopulations were generated for each data set in order to assign equal sample sizes per condition.

Single Factor ANOVA tests also calculated variation in survival rate and emergence of winged morph rate between diet treatment/rearing condition.

All sample sizes and critical values for statistical tests can be found in the Appendix A (Appendix A).

### 2.6. Fitness and Morphometric Measurements

Reproductive female clones were allowed to lay nymphs on a 4–6 whorl stage *V. faba* plant for a 24 h or less period of time. Adult aphids were then removed, and the nymphs were allowed to develop on the plant for “X” additional hours. Varying “X” hours allowed for nymphs to be aged to various developmental stages (1st, 2nd, 3rd, 4th instar nymph, young adult, reproductive adult). Once aged to the correct developmental stage, 8 individual nymphs were randomly removed from the plant and were immobilized on ice. These 8 individuals were weighed (weight is commonly used as a surrogate fitness measurement, [50]) using a New Classic MF Semi-micro Analytical balance (Metter Toledo, LLC, Columbus, OH, USA) and their mass was recorded in mg. Once weighed the individuals were transferred to individual wells on a 24-well plate and transferred to a dissecting stereomicroscope (Leica Microsystems, Wetzlar, Germany) where the individual’s body length (anterior most point of the head to posterior most point of the cauda) and siphunculus length were measured at 5× magnification using the on-lens ruler. This measurement was then converted to length in millimeters. This was repeated three times, so that a total of 24 individuals, from three independent blocks (technical replicates) were measured from each aphid sub-line (biological replicates) at each developmental time point.

## 3. Results

### 3.1. Dietary Effects on the Timing of Developmental Events

First instar aphids (0–8 h; 4.0 ± 4.0 h), viviposited by the same clonal mothers on a *V. faba* plant were divided into three diet treatments (Plant, Leaf, Diet) and were allowed to develop until reproductive maturity under one of the three randomly assigned rearing conditions. This experiment was simultaneously carried out with three biological replicates from three independent sub-lines (Fava 1, 2, 3).

On average aphids reared on whole plants completed their first molt (transition from 1st to 2nd instar) approximately 41.6 h after they were viviposited (Standard Deviation (SD): 3.78, Range (R): 32–51 h). The average time for the remaining molts were as follows; note that times are continuous from viviposit to each developmental event and are not reset at each event (Figure 2, green): second molt ~81.7 h (SD: 6.02, R: 72–97), third molt ~126.7 h (SD: 7.32, R: 113–148), and fourth molt ~190.1 h after viviposition (SD: 11.15, R: 168–226). Whole plant-reared aphids became reproductive and viviposited their first offspring approximately 235.4 h after their own viviposition (SD: 14.85, R: 210–279). Overall, the three independent sub-lines yielded very similar rates of development, however, a sub-line effect was observed for Fava 2 during certain developmental stages (Appendix A).

Leaf-reared aphids survived to the reproductive adult stage at a similar rate compared to Plant-reared aphids (78.5% vs. 81.3%, respectively, survived to this life stage; Appendix A). The emergence of the winged morph recorded at the adult stage also occurred at a similar rate between the whole plant and leaf treatments (34.6% vs. 27.8% developed wings; Appendix A). However, in contrast to the whole plant treatment, Leaf-reared aphids experienced a developmental delay (Figure 2, yellow). On average the time to complete each developmental event was: first molt ~47.1 h (SD: 6.49, R: 34–59), second molt ~88.6 h (SD: 6.24, R: 79–105), third molt ~135.3 h (SD: 7.94, R: 119–157), fourth molt ~206.8 h (SD: 10.94, R: 184–232), and nymph laying ~277.4 h (SD: 19.96, R: 230–319) post viviposit. Each event, respectively, took ~5.5, ~6.9, ~8.6, ~16.8, and ~42.0 h longer on average compared to Plant-reared aphids. This difference in developmental timing was found to be statistically significant for all life stages (Appendix A). No sub-line effect was observed for any of the developmental stages for the Leaf treatment (Appendix A).

Initially, Diet-reared aphids performed poorly, and zero aphids survived past the 3rd instar (vs. 81.3% on Plant, Appendix A). In fact, out of the 112 aphids (combined from 3 sub-lines and 2 replicates), 74.1% died as 1st instar nymphs never completing a molting event, 18.8% died as 2nd instar nymphs, and only 7.1% made it to the 3rd instar. Based on our optimization trials (see methods) survival rates greatly increased if aphids were allowed to remain on the whole plant until the 2nd instar stage. Second instar nymphs were then transferred to the Diet treatment at that life stage (see methods). Diet-rearing starting at the 2nd instar yielded a survival rate that was higher, but still significantly lower compared to Plant-reared aphids (53.9% vs. 81.3% survival respectively; Appendix A). Unexpectedly, only one diet-reared aphid (2.1%) developed into a winged morph, whereas both the Plant- and Leaf-reared aphids developed into winged morphs approximately 30% of the time.

Similar to Leaf-rearing, Diet-reared aphids also experienced a developmental delay compared to aphids that were reared on the whole Plant (Figure 2, blue). On average the time to complete each developmental event was: first molt ~42.7 h (SD: 4.48, R: 35–52), second molt ~92.3 h (SD: 10.61, R: 75–120), third molt ~144.1 h (SD: 14.46, R: 120–184), and fourth molt ~213.1 h (SD: 18.92, R: 168–263), and nymph laying ~274.7 h (SD: 29.50, R: 216–407) post viviposit. Each event, respectively, took ~1.1, ~10.6, ~17.4, ~23.1, and ~39.3 h longer on average than the time required for aphids reared on whole plants. This difference in developmental timing was found to be statistically significant for all life stages (Appendix A). In addition to the delay in the average timing of developmental events, there also appears to be much more variation in the timing of these developmental events. Specifically, the range between the minimum and maximum value and the standard deviation is ~2 fold larger for most Diet-reared stages (excluding the first molt, which was reared on a whole plant) compared to the corresponding Plant- and Leaf-reared stages. A significant sub-line effect was only observed for sub-line Fava 3 during the third molt stage (Appendix A).

### 3.2. Effects of Plant Age and Transgenerational Stress on the Timing of Developmental Events

Aphids reared on older *V. faba* plants (~3.5 weeks and 7−8 whorls vs. 2.5 weeks and 4−5 whorls) or whose grandmothers/great-grandmothers were exposed to stressful conditions during development (overcrowding) have a similar survival rate compared to aphids reared on whole plants with 4−5 whorls without transgenerational stress (80.0%, 75.0% vs. 81.3%, respectively; Appendix A). Moreover, aphids reared on older plants and those exposed to overcrowding emerge as winged morphed adults at a similar rate as aphids reared on younger plants and under less crowded conditions (32.0%, 16.7%, vs. 27.8%, respectively; Appendix A) (from here on referred to as Old, Stressed, and WT, respectively). However, similar to aphids reared on the Leaf or Diet treatment (above), it appears that aphids reared under Old and Stressed conditions experienced a developmental delay compared to WT, though to a lesser extent (Figure 3). The developmental timing of the 1st and 2nd molt was slightly longer, but not statistically different from WT-reared aphids (1st: WT = 41.1, Old = 43.1, Stressed = 41.1; 2nd: WT = 82.6, Old = 85.7, Stressed = 81.4, Appendix A). When observing the timing of the 3rd and the 4th molting events, the Stressed aphid population is not significantly different from the WT population, however aphids reared on the Older plant treatment experienced a significant delay in development for the latter molts (3rd: WT = 127.6, Old = 136.9, Stressed = 128.2; 4th: WT = 189.3, Old = 205.6, Stressed = 191.4, Appendix A). Also, the length of time until reproductive maturity was significantly delayed for aphids reared on both the Old plant treatment and the Stressed conditions treatment compared to WT aphids (WT = 234.3, Old = 251.1, Stressed = 253.8, Appendix A).

### 3.3. Effects of Wing Morph on the Timing of Developmental Events

In this study, no diet treatment or rearing condition was associated with a higher proportion of winged to non-winged morph adults. Most rearing conditions (Leaf, Old, Stressed) led to a similar proportion, or a lower proportion (Diet) of winged to non-winged morph adults as observed after Plant rearing or under WT conditions (see above). When comparing the development rates between winged and non-winged morphs, winged aphids took significantly longer to lay nymphs compared to non-winged aphids that were reared together, regardless of what diet treatment or rearing condition they were exposed to (Figure 4).

First, comparing the developmental timing of reproductive maturity of non-winged individuals versus winged individuals from both Plant-reared and Leaf-reared treatments (Diet-reared were excluded because only one winged morph individual was observed) there was a significant difference between morphs (wing vs non-) and diet treatment (Plant vs Leaf) (Appendix A). The interaction between these factors (morph and diet treatment) was not significant. In sum, these data suggest that both morph and diet treatment have a significant effect on developmental timing, but that the effects are independent of each other. The diet treatment (Leaf) appears to have a larger developmental effect, increasing the time it takes non-winged females and winged females to viviposit by 19.4% and 15.6%, respectively (compared to Plant), while morph (winged morph) only increases this time 10.9% and 7.4% in Plant- and Leaf-reared aphids, respectively (compared to non-wing morph).

Second, comparing the developmental timing of reproductive maturity of non-winged individuals versus winged individuals from WT and Stressed rearing conditions (Old Plant data not available) there was both a significant difference between morphs (winged vs. non) and rearing conditions (WT vs. Stressed) (Appendix A). The interaction between these factors (morph and diet treatment) was not significant. These data again suggest that both morph and rearing background have a significant effect on developmental timing, but that the effects are independent of each other. In this case, the winged morph appears to have a slightly larger developmental effect, increasing the time it takes WT females and Stressed females to viviposit by 9.6% and 8.9%, respectively (compared to non-wing morph), while rearing background (Stressed) only increases the time to viviposit by 6.8% and 6.0% in non-winged and winged aphids, respectively (compared to WT).

### 3.4. Fitness and Morphometric Measurements at Each Developmental Stage

Whole plant-reared aphids were collected, and fitness and morphological data were measured at each nymphal developmental stage and as pre-reproductive adults (developmental times observed above, and molt counting were used to determine developmental age). On average 1st instar nymphs weighed 0.15 mg and were 1.14 mm long, 2nd instar nymphs weighed 0.34 mg and were 1.54 mm long, 3rd instar nymphs weighed 0.77 mg and were 2.03 mm long, 4th instar nymphs weighed 1.62 mg and were 2.84 mm long, and young, pre-reproductive adults weighed 2.21 mg and were 3.34 mm long. Overall these two measurements increased at a similar constant rate across development, as seen by their linear-like relationship (Figure 5A). While on a population level there was a significant difference in both body mass and length (ANOVA: *p* < 0.05 for all comparisons). Specifically, measurements between individual aphids varied greatly, and at times overlapped with measurements from different developmental stages. Therefore, body mass and/or length should not be used to determine an individual’s developmental age (Figure 5A).

Another measurement that was taken was the length of the pheromone producing siphunculus. On average, 1st instar nymph siphunculi were 0.17 mm long, 2nd were 0.30 mm long, 3rd were 0.43 mm long, 4th were 0.70 mm long, and young, pre-reproductive adult siphunculi were 0.88 mm long. While body length and mass varied greatly between individuals, the length of this appendage was extremely consistent between aphids within a single developmental stage (Figure 5B−C). Siphunculus lengths for each developmental stage were significantly different from sequential stages (ANOVA: *p* < 0.05 for all comparisons), and there was no overlap in measurements between sequential stages. These results suggest that siphunculus length is an excellent measurement to use as a no-kill tool and proxy to predict the nymphal instar of *A. pisum* developing on whole plants.

## 4. Discussion

In this study we show for the first time how nymphal development of *A. pisum* varies when they are reared on different experimental diets, such as artificial diet, isolated leaf samples and whole plants. While it has been shown previously that aphids could be reared from young nymphs to reproductive adults on either isolated leaf samples or artificial liquid diets [39,45,48], this study reveals that there are significant differences in aphid development rates depending on which diet the aphid is reared upon. For example, the nymphal developmental period for each instar and the time it takes for the aphid to reach reproductive maturity is significantly longer when reared on an isolated leaf sample or on an artificial diet when compared to a whole host-plant. This developmental delay was not isolated to one developmental stage but was observed across pre-reproductive development. Additionally, the delay was not uniformly distributed across the developmental period, but instead was asymmetric with various stages experiencing delays to a greater or lesser extent (Figure 2).

In this study, we also showed that the variation between and within biological replicates for each treatment (intra-treatment variation) was higher than we initially expected in all three diet treatments. Aphid mothers were allowed to viviposit for 8 h; therefore, we expected the variation in the observed timing of developmental events to approximately fall within an 8 h window. However, in this study we observed ~2.5× the amount of variation in the timing of the first molt and up to ~8.5× the amount of variation for the timing of nymph laying for aphids that were reared on whole plants or on isolated leaf samples, compared to variation due to viviposit time (8 h). Additionally, we observed an additional 2-fold increase in the variation for Diet-reared aphids compared to the whole plant or isolated leaf sample treatments.

Understanding how aphid developmental rates can vary dramatically depending on the diet they feed on is important for future studies to consider when interpreting data from experiments where aphids are reared on different artificial and/or natural diet conditions. For example, even though whole plants are thought to vary widely in nutritional profiles both between individual plants and within parts of the same plant [54], we observed the least amount of variation in the timing of aphid developmental events within and between biological replicates when reared on whole plants of the same age compared to aphids reared on a nutritionally controlled, uniform artificial diet treatment (Figure 2). In consequence, if the life stage of individual aphids is not accounted for in artificial diet experiments, variation observed in the measured dependent variables may heavily be influenced by differences due to the variation in aphid developmental life stages and not necessarily the dependent variable itself.

### 4.1. Possible Sources of Inter- and Intra-Treatment Variation

In contrast to the results we observed in this study, we expected to see the largest degree of intra-treatment variation in the whole plant treatment. For example, nutrient levels vary if an aphid feeds from leaves, buds, or stems (plant architecture is known to affect nutrient transport), and therefore aphids would have to feed for longer or shorter amounts of time to get enough energy to molt depending on the plant part they feed upon [55,56]. Batch effects (differences from plant to plant) and sub-line specific variation also can add more variation to developmental rates. Additionally, aphids are known to have a behavioral alarm response to plant vibrations causing them to drop off of the plant [57,58,59,60,61]. Plant handling during molt checking (while done delicately), potentially could have triggered this response, which again could unevenly add time to the development of individuals.

Aphids reared on non-whole plants are fed a much more consistent diet than plant-reared aphids. Isolated leaf samples eliminate plant architecture variation, and metabolites within the artificial diet were standardized for the Diet treatment. Additionally, the much smaller size of the Leaf- and Diet-rearing arenas, limited the aphid’s food source search area. We expected the combination of diet composition uniformity and search area reduction to result in Leaf-reared aphids exhibiting less intra-treatment variation compared to Plant-reared aphids, and Diet-reared aphids exhibiting the least variation of all three treatments. However, in contrast to the expected outcome, we observed a similar level of intra-treatment variation in Leaf-reared aphids, and ~2× more variation in Diet-reared aphids compared to aphids reared on whole plants.

While aphids have been shown to prefer to reside on leaves (especially buds) over stems, Legrand and Barbosa (2000) demonstrated that plant morphology and within plant distribution had no effects on growth and fecundity, as the aphids are most likely to feed and lay nymphs wherever they initially settle [36,62]. The lack of an effect due to plant morphology, and the lack of a correlation between intra-treatment variation and diet uniformity suggests that differences are not due to the chemical composition of the diet varying between whole plant structures.

Possible reasons for the observed inter-treatment developmental delay and/or the increased amount of intra-treatment variation are that the isolated leaf samples may be producing altered nutritional profiles and/or defense responses compared to whole plant samples. Also, the leaf samples and the artificial diet treatment cannot replicate the turgor pressure present in whole plant cells, and/or they cannot produce some unknown chemical and/or tactile phagostimulation agent [27,39,63]. All of these factors could affect the overall feeding rate of the aphids. While we know that the aphids in these experiments were not dying from starvation (observance of food dye in both body and honeydew), one or more of the above factors could lead to aphids feeding at a reduced rate compared to whole plant reared aphids. This lower feeding rate could therefore require aphids to feed for longer periods of time to acquire sufficient amounts of nutrients/energy needed for each developmental stage. It has been shown previously that the green peach aphid (*Myzus persicae*) ingests twice as much sap from a host plant seedling than an artificial diet sachet, and as a result grows to approximately only half the size of plant sap reared aphids [64]. Additionally, from observing stylet paths, it has been demonstrated that an aphid will probe several sieve tubes before it feeds [65]. These two experiments both indicate that the aphid can discriminate between food sources and will actively feed more on a preferred source. After finding an ideal food source, mechanical limitations may prevent feeding at an ideal rate. Isolated leaf samples no longer have the natural turgor pressure established by the vascular system of an intact plant and the degree to which the parafilm is stretched over the Diet-rearing arena may either prevent stylet penetration all together or result in a non-optimal flow rate once pierced, all affecting feeding rates in mechanical ways. Finally, feeding rate may be limited for behavioral reasons. It has been shown that the presence of sucrose can be phagostimulatory and promote feeding in otherwise identical diets [46]. Certain amino acids, such as Methionine have been shown to also promote feeding behaviors [39,50]. Additionally, there could be an odorant, pheromone, or allelochemical that is released by the plant, or a structure on the plants surface that similarly promotes feeding behaviors [66], and when absent in non-whole plant diets leads to a diminished feeding rate, decreased fitness, and/or a delayed development.

### 4.2. Developmental Effects of Plant Age, Transgenerational Responses to Environmental Cues, and Morphotype

This study shows that aphids reared on an older whole plant, also experiences a delay in the timing of developmental events. No differences were observed during the first two molts, but from the third molt on, old-Plant-reared aphids experienced a delay in development similar to that seen on Leaf-reared aphids. Potentially, as the plant ages, less nutrients are available to the aphid resulting in a delay during later developmental stages. In this study, the plant still appeared healthy at the completion of trials, so the delay cannot be ascribed to low plant vigor, but instead maybe due to some physical or chemical change in the plant once it reaches a certain age. It has been shown that older black spruce needles have lower levels of both nitrogen and phosphorous, lower photosynthetic rates, and lower nutrient use efficiencies than younger needles [67]. Similarly, there is a decline in nitrogen uptake, in nitrogen concentration, and in respiration within grape roots as they age [68]. Finally, it has been demonstrated that as *Arabidopsis* ages there is a decay in JA response, the phytohormone responsible for insect resistance [69]. Together, these three studies demonstrate that the chemical and molecular make up of a plant changes with age, and perhaps a change in plant defenses, nutrients, and/or hormones during plant development may contribute to these observed aphid phenotypes.

In this study we also observed that environmental cues, resulting from aphids reared in a stressful environment with overcrowding and feeding on a host plant displaying low vigor, could be passed on from generation to generation, and that these maternally inherited signals could influence development. This is most commonly shown in aphids, by the induction of winged (alate morph) females. It has previously been shown that exposure to overcrowding and/or poor-quality sap can be sensed by a mother aphid (alarm pheromone and tactile stimulation), who then triggers a lower secretion of ecdysone in her ovary, signaling the formation of winged morph daughters in next generation (likely through maternal factors and/or epigenetic changes); allowing her offspring a means to escape from the non-ideal conditions the mother aphid is currently facing [4,5,6,12,13]. Given the telescopic development of aphids (granddaughter present as an embryo nested within a developing daughter nested inside the adult mother), it is of interest if these environmental cues perceived by the mother can also be inherited by the granddaughter. Previous work has shown that environmentally induced-winged daughters, cannot themselves react to the environment, and will lay non-winged offspring, regardless of any environmental cues they may experience as an adult [4,13]. So, if stressors processed by a mother can be passed not only to the daughter, but also to her granddaughter, we would not expect to see this through a proxy like wing development, but perhaps the effects of this transgenerational stress can be observed in another way, such as developmental timing.

In this study, we demonstrate that cues triggered by transgenerational stress can be inherited by the granddaughter and can result in developmental defects such as a delay in the timing of events such as nymph laying. During each trial in this study, it was noted if an adult was winged or non-winged when recording the time, she laid her first offspring. When comparing the developmental time of winged versus non-winged females reared under the same condition it was observed that independent of the diet treatment and/or rearing condition, winged females always took significantly longer to lay nymphs than their non-winged counterparts. However, our data also suggests that transgenerational stress can also lead to a delay in development, independent of the delay caused by winged females taking longer to develop. While “Stressed” granddaughters took a similar amount of time to complete nymphal development as “WT” granddaughters, they took significantly longer to lay nymphs. In this study, the induction of winged females was actually lower in aphids whose grandparents were exposed to stress than those not, so the observed delay in nymph laying cannot be explained by a higher proportion of the slower developing winged morphs. Thus, it is possible that while small, the response triggered by past stressors is still inherited by the granddaughters, perhaps through epigenetic changes (JH binding protein gene is highly methylated in winged offspring [4]), and these changes result in physical or behavioral changes that prolong development (reproductive maturity) of both alate and apterous females.

### 4.3. Siphunculus Length: A Proxy for Determining Developmental Age

In this study we validated that siphunculus length can be used as a proxy/surrogate to accurately determine an individual’s developmental stage, so that molting does not have to be observed for every individual. Here, all aphids of a particular stage had siphunculi of a significantly similar length and then there was a large, instantaneous growth in length at each sequential molting event (Figure 5). Similarly, previous studies have found that the antennal length may also be used to precisely age *A. pisum* nymphs [41,70,71]; however, those measurements were not examined in this study.

## 5. Conclusions

In summary, this study characterized the nymphal and pre-reproductive developmental period of the commonly used laboratory strain of *A. pisum*, LSR1, as it develops on it universal host-plant (*V. faba*), common non-natural diets (isolated leaf samples and artificial liquid diets), and under sup-optimal rearing conditions that could potentially occur (older host plant, trans-generationally inherited stress from overcrowding and low host plant vigor). Unexpected levels of intra-treatment variation in the timing of these developmental events was observed in all diet treatments, as was significant inter-treatment variation, leading to strong developmental delays when feeding on non-natural diets and weaker delays when reared under sub-optimal conditions (older plant, stressful conditions). Therefore, if not tightly controlled, commonly used experimental conditions such as diet choice, diet quality, and the environment of the mother and grandmother can influence the timing of these developmental events, necessitating that these artificially caused fluctuations must be accounted for before determining the effects caused by other experimental variables. Finally, given that molting and nymph laying time frames are dependent on diet and environmental rearing conditions, additional confirmations (molt counting, siphunculus measuring) are vital if accurate developmental staging is required or important in the experimental design.

## Figures and Tables

**Figure 1 insects-10-00260-f001:**
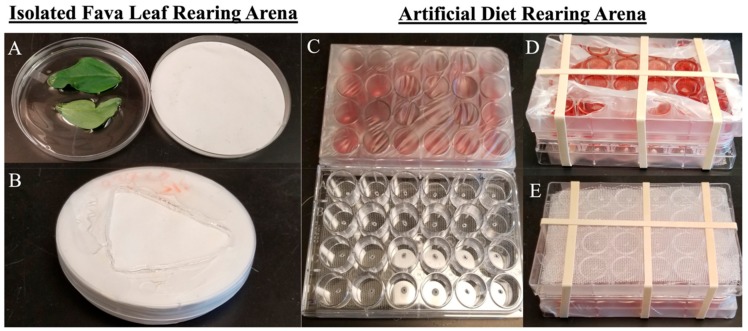
Diet Treatment Rearing Arenas. (**A**) Isolated Fava Leaf Diet Treatment (Leaf) Rearing Arena (open orientation); (**B**) Isolated Fava Leaf Diet Treatment (Leaf) Rearing Arena (closed orientation, top view); (**C**) Artificial Diet Treatment (Diet) Rearing Arena (open orientation); (**D**) Artificial Diet Treatment (Diet) Rearing Arena (closed orientation, top view); (**E**) Artificial Diet Treatment (Diet) Rearing Arena (closed orientation, bottom view).

**Figure 2 insects-10-00260-f002:**
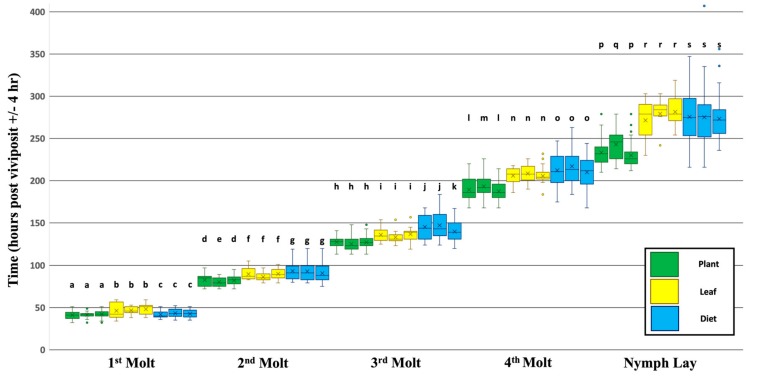
Dietary Effects on the Timing of Developmental Events. The timing of each developmental event (X axis labels) is portrayed as three box plots (biological reps) per each diet treatment (green: plant, yellow: leaf, blue: diet). Time is shown as hours post viviposit and does not reset after each event. Box plots represent the distribution and variation in the observed timing of each event (horizontal line: median value, x: mean value, bars: minimum and maximum values, box: range from lower to upper quartile, circles: outlier values). Letters above box plots signify statistical significance between biological replicates and between diet treatments. Initial sample sizes Treatment (Bio. Rep 1, 2, 3): Plant N = (65, 80, 61), Leaf N = (20, 17, 20), Diet N = (38, 70, 104). See Appendix A for more details.

**Figure 3 insects-10-00260-f003:**
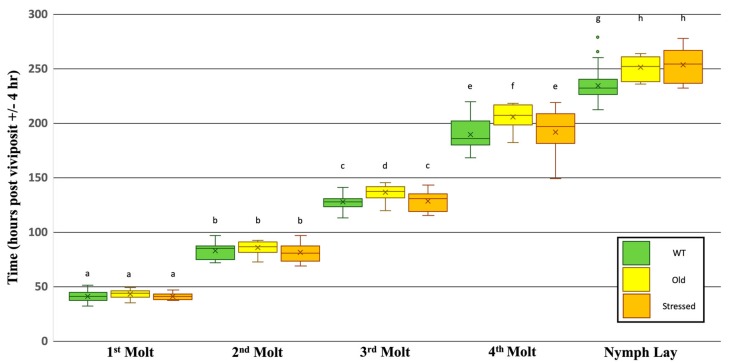
Effects of Plant Age and Transgenerational Stress on the Timing of Developmental Events. The timing of each developmental event (X axis labels) is portrayed as one box plot per each set of rearing conditions (green: WT (4–5 whorl plant, no transgenerational stress), yellow: Old (7–8 whorl plant, no transgenerational stress), orange: Stressed (4−5 whorl plant, transgenerational stress). Time is shown as hours post viviposit and does not reset after each event. Box plots represent the distribution and variation in the observed timing of each event (horizontal line: median value, x: mean value, bars: minimum and maximum values, box: range from lower to upper quartile, circles: outlier values). Letters above box plots signify statistical significance between rearing conditions. Initial sample sizes WT N = 65, Old N = 25, Stressed N = 20. See Appendix A for more details.

**Figure 4 insects-10-00260-f004:**
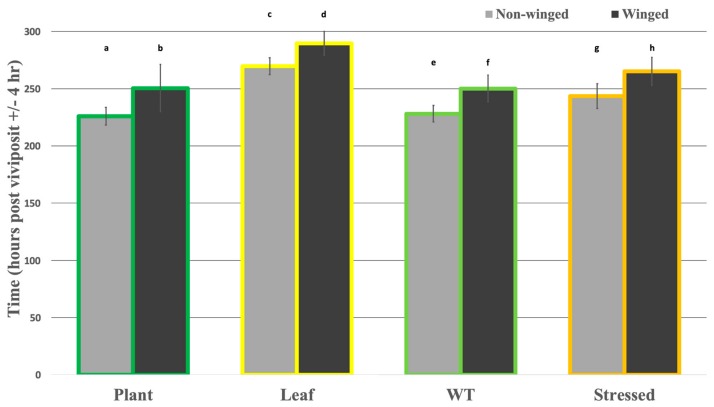
Effects of Diet Treatment or Rearing Condition versus Wing Morph on Average Nymph Laying Time. The average time for an aphid to lay her first nymph is portrayed as four pairs of bar graphs. Each pair represents a different diet treatment or rearing conditions the aphids were exposed to (Outline color: green: Plant, yellow: Leaf, lime: WT (4–5 whorl plant, no transgenerational stress), orange: Stressed (4–5 whorl plant, transgenerational stress)), and each bar within the pair represents the wing morph of the aphids observed vivipositing nymphs (Bar color: light gray: non-winged, dark gray: winged). Letters above bars signify statistical significance between wing morph and diet treatments (plant vs. leaf) and between wing morph and rearing conditions (WT vs. Stressed).

**Figure 5 insects-10-00260-f005:**
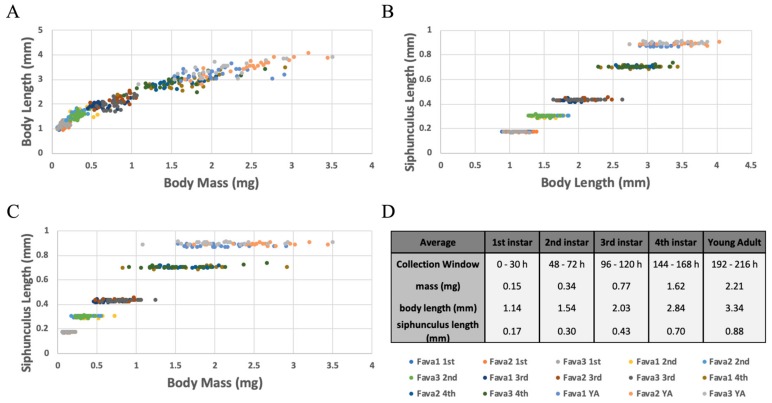
Comparisons of Fitness and Morphometric Measurements at Each Developmental Stage. (**A**–**D**) Aphids reared on whole Fava plants for X−Y hours were collected and three measurements were taken per aphid. For each life stage, 8 aphids were measured from each of three biological replicates (Fava 1, 2, 3). The color of each data point corresponds to the aphid’s life stage and biological replicate. A color key is included below panel D. Measurements taken were then compared pairwise (**A**–**C**) or averaged (**D**). (**A**) Body Length versus Body Mass; (**B**) Siphunculus Length versus Body Length; (**C**) Siphunculus Length versus Body Mass; (**D**) Collection window and average body mass, body length, and siphunculus length for each corresponding life stage.

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
