# Peer review of "The Effects of Different Diets and Transgenerational Stress on Acyrthosiphon pisum Development"

_insects, 2019, doi:10.3390/insects10090260_

Round 1

Reviewer 1 Report

The merit of the MS is quite interesting and may be of high importance not only for researchers dealing with experimental studies on A. pisum but I also found it interesting in a few aspects concerning possible evolutionary mechanisms in aphids in general. Especially in terms of epigenetic factors, which are however only superficially treated in the MS. I have only a few suggestions for authors:

1) if you do not discuss or comment the results of morphometry why mention it at all in the MS? There is so many detailed data that in my opinion if you exclude this only minor part of research the remaining part would be more comprehensive. The weight data could stay but the measurements (which I doubt their credibility in macroscopic measuring and not on mounted specimen on microscopic slide) put nothing new and important in this MS. Please consider removing this small part from the MS.

2) when you measure time - how much time took the experimental activities - removing aphids, transferring them, closing the experimental 'capsules'. Was this time taken into account in the experiment and the final results?

3) when you refer to escape behavior of aphids, where there any thanatotic behaviors involved (such as in Bilska et al. 2018) - they may last a while in non-myrmecophilous aphids and may also affect the results.

Also, in Tables (Appendix A) the numbers should be either adjusted to right margin and given additional zeros where needed (e.g. 0.02000) or to the left, if no additional zeros ar going to be added. In the centred position and with various precision it seems to me mathematically unjustified, unprofessional and not clear. 

Reviewer 2 Report

The submitted paper ‘The effects of different diets and transgenerational stress on Acyrthosiphon pisum development’ by Daniel Pers and Allison K. Hansen is a very interesting study that will be of use to all aphid researchers dealing with experimental biology. Indeed, there was a need for a comparative study showing in detail the significance of pre-experimental treatments of aphids in the interpretation of the results, especially when aphid life parameters are studied. Also, the finding that there is a lower intra-variation of aphid life parameters on whole plants as compared to leaf discs or artificial diets, despite the hard to define variation in internal plant environment, is very important for future researchers of aphid responses to pesticides or natural xenobiotics. The experiments were very well designed and carefully performed. I expect that the article, when published, will be highly cited.

There are minor points for consideration:

Figures and Tables: Please, provide the number of replicates (aphids) used in each statistical analysis, e.g. Fig 2. plant: n=xx, leaf: n=yy, diet: n=zz, and likewise in other figures (WT, old, stressed, etc.). It is especially important in the case of diet-reared aphids where the mortality was high. In Material and Methods, only approximate numbers of replicates are given. In the section devoted to statistical analysis, there is information that outlier data points were removed. So, what was exactly the number of replicates analyzed? Please give information on the statistical test used for the displayed results.

Figs. 3 and 4. Please, define WT, old, and stressed in the caption to the figure. What exactly does ‘old’ mean? There is some description in Material and Methods and in the Results but it should be more precise.

17: ‘anatomical method’ – I will argue that the measurements of external body elements should me named ‘morphological method’; ‘anatomy’ is in fact ‘internal morphology’ 117-118: Instead of ‘the number of nymphs observed giving birth’ it should probably be: ‘the number of nymphs laid’ 118: ‘the time (hours post viviposit) were recorded’ – what was the purpose of that if the reproducing female was removed from the experiment, or should it be: ‘hours pre viviposit’? 131: mistake in temperature, it certainly was not 80ºC; was it really 20% RH? I guess, the two values were misplaced. 135: embed or embedded? 190: imprecise expression; ‘Once these nymphs became reproductive’; it should be: ‘Once these nymphs became reproductive adults’

Reviewer 3 Report

General comments:

As most previous work on aphid development is mainly focusing on wing polymorphism, sexual and asexual reproduction, symbiosis of the adult or embryonic stages, little is known about the aphid nymphal development. Using the model pea aphid, Acyrthosiphon pisum and its commonly used host plant Vicia faba, this study compared the impact of different diets as well as plant ages and transgenerational stressors on aphid nymphal development. The results showed a significant delay in nymphal and pre-reproductive adult development on isolated leaves and artificial diets compared to intact host plants. It is useful to have a direct comparison of aphid development on commonly used diets to remind the aphid field to carefully design experiments and interpret results in the context of different conditions. In addition, the authors found that the stresses aphids experienced can influence nymphal development in the following generation/s. Lastly, it is very interesting that they found the appendage, siphunculus, as a reliable nymphal stage indicator. Although this may be A. pisum specific, this finding will be a helpful reference for the aphid researches concerning specifically different nymphal stages. 

Below are some minor issues:

For the trials on different diets, it is very nice to have three aphid sub-lines as biological replicates, which is much nicer than simply technical replicates. However, it is not clear how many times the experiment was replicated?   Figure presentation needs improvement. The labels for the y-axis of figures 2, 3, 4 are too small to read and the lines for the axes are missing or too faint to be seen. The error bar in figure 4 can be made clearer. The statistical label in figure 4 is confusing (a, b, …, h), it is not clear what has been compared. Line 135: In the diet treatment experiment, please clarify where those leaves were isolated from the plants in terms of position, e.g. top younger leaves or bottom older leaves? Line 182: “unfavorable conditions” please clarify how these conditions were generated. Line396: keyword is missing? “…were significant ? from sequential stages…”
